# Ursolic Acid Analogs as Potential Therapeutics for Cancer

**DOI:** 10.3390/molecules27248981

**Published:** 2022-12-16

**Authors:** Siva S. Panda, Muthusamy Thangaraju, Bal L. Lokeshwar

**Affiliations:** 1Department of Chemistry and Physics, Augusta University, Augusta, GA 30912, USA; 2Department of Biochemistry and Molecular Biology, Augusta University, Augusta, GA 30912, USA; 3Georgia Cancer Center, Augusta University, Augusta, GA 30912, USA; 4Research Service, Charlie Norwood Veterans Administration Medical Center, Augusta, GA 30912, USA

**Keywords:** ursolic acid, anticancer, antitumor, synthesis, modifications, cell lines

## Abstract

Ursolic acid (UA) is a pentacyclic triterpene isolated from a large variety of vegetables, fruits and many traditional medicinal plants. It is a structural isomer of Oleanolic Acid. The medicinal application of UA has been explored extensively over the last two decades. The diverse pharmacological properties of UA include anti-inflammatory, antimicrobial, antiviral, antioxidant, anti-proliferative, etc. Especially, UA holds a promising position, potentially, as a cancer preventive and therapeutic agent due to its relatively non-toxic properties against normal cells but its antioxidant and antiproliferative activities against cancer cells. Cell culture studies have shown interference of UA with multiple pharmacological and molecular targets that play a critical role in many cells signaling pathways. Although UA is considered a privileged natural product, its clinical applications are limited due to its low absorption through the gastro-intestinal track and rapid elimination. The low bioavailability of UA limits its use as a therapeutic drug. To overcome these drawbacks and utilize the importance of the scaffold, many researchers have been engaged in designing and developing synthetic analogs of UA via structural modifications. This present review summarizes the synthetic UA analogs and their cytotoxic antiproliferative properties reported in the last two decades.

## 1. Introduction

Cancer is the first or the second leading cause of death worldwide, accounting for 19.2 million new cases and 9.6 million deaths, in 2020 [1]. The most common causes of cancer death in 2020 were lung, colorectal, liver, stomach, and breast cancer. In addition, each year more than 40,000 children develop cancer [1]. In the United States, there will be an estimated 1.9 million new cancer cases and 609,360 cancer deaths in 2022 [2].

The cancer burden continues to grow globally, exerting tremendous physical, emotional, and financial strain on individuals, families, communities, and even the national health systems. Compared to developed counties, low- and middle-income countries have low survival rates because of the lack of accessible early detection, quality diagnosis, treatment, and survivorship care. The global economic burden of cancer is unknown, as accurate economic data are not available from many countries [1]. In the United States, the national patient economic burden associated with cancer care was USD 21.09 billion in 2019 [3].

Current treatments for malignant cancers include surgery, radiation, chemotherapy, adjuvant therapy, hormone therapy, and immunotherapy. Despite all these therapies and the tremendous progress, refinement, and entry of novel drugs, procedures, and treatment options, cancer continues to be a leading cause of death. The challenges to reducing cancer burden, and specifically for treatment failures, are associated with the development of drug resistance, disease progression, and dose-limiting systemic toxicities of potent drugs [4]. The much-heralded immunotherapy has low efficacy, with less than 20% of patients responding to the treatment. Novel effective and efficient pharmacological agents that act through unconventional mechanisms to enhance existing therapies or kill tumor cells resistant to other existing therapies are urgently needed.

Over four decades, bioactive compounds isolated from natural sources, such as plant and marine organisms, the natural products, have dominated the cancer prevention, treatment, and drug development areas. Natural products (NP) are one of the main sources of diverse pharmacologically active compounds [5,6,7,8,9,10,11]. NPs and NP-scaffolds comprise a large portion of current-day pharmaceutical agents (over 70% of FDA-approved drugs) [12,13]. At present, several native or modified NPs have attained the status of cancer therapeutics. These include irinotecan, vincristine, etoposide, and paclitaxel from plants, actinomycin D, and mitomycin C from bacteria, as well as marine-derived bleomycin.

## 2. Triterpenes as Bioactive NPs

*Triterpenoids* are compounds with a carbon skeleton based on six isoprene units, metabolites of isopentenyl pyrophosphate oligomers, representing the largest group of phytochemicals. These are a large and structurally diverse group of NPs having about 200 distinct skeletons, and more than 20,000 triterpenoids are isolated and identified from nature [14,15].

In the 1920s, for the first time, Ursolic acid (UA) was isolated and identified from epicuticular waxes of apples. UA is one of the most abundant and well-studied triterpenoids found in various cuticular waxes of edible fruits (apple, blueberry, cherry, cranberry, Japanese loquat, lemon, olive, orange, peach, pear, prune, quince, tangerine, and tembusu), leaves (coffee, elder, glossy privet, hawthorn, lavender, nerium, marjoram, olive, organum, rosemary, thyme, and whorled rosinweed), flowers (loquat, marigold), and bark (elder, olive, and silver birch) of medicinal plants [16,17,18,19,20]. UA is present in most edible plant products.

UA holds an important place among various triterpenoids because of its wide range of biological activities. UA is abundantly present in medicinal plants and shows many pharmacological activities including antiproliferative [21,22,23], antimicrobial [24,25], antiviral [26,27], antioxidant [28,29,30], and anti-inflammatory activities [30,31].

UA has been extensively applied in cancer treatment in Traditional Chinese Medicine (TCM) for several years. Over the last two decades, UA has been tested on several malignant cancers for both the prevention of cancer progression and as a treatment. Most studies are limited to its activities against energy metabolism, cell proliferation, and antioxidant activities [32,33,34,35,36,37,38,39,40,41,42,43]. These studies have established at least in vitro, that UA inhibits many activities of cancer cell growth, energy utilization, and tumor cell-induced inflammation as summarized in Figure 1 [44,45,46,47,48,49,50,51,52,53].

Despite its versatile functions, UA has limitations. Most studies on UA are limited to cell cultures in vitro where UA has access to cells and it is not a concern. UA diluted in biocompatible solvents are added to cell cultures, and its activities on cells are determined by many analytical techniques. However, the availability of UA at the site of the tumors in an intact body is a significant concern. Like many natural, medicinal compounds, administration of UA parenterally is seldom used due to its poor solubility and potential toxicity of the solvents, such as Dimethyl Sulfoxide or Dimethyl Formamide. The oral route, via its addition to diets or drinking water, is of little use. The bioavailability of UA, in plasma, is limited to ≤500 nM [54]. Most in vitro studies have shown the inhibition of tumor cell proliferation in low micromolar concentrations. Studies using transgenic prostate cancer models have reported good tumor control within 12 weeks of treatment in well-established transgenic tumors via a solid diet at 1% *w*/*w*. The authors reported a ~60% reduction in prostate tumor volume following a 12-week observation, indicating the high efficacy of UA in the prevention of tumor progression in this model [55]. Studies in human volunteers were less promising, concerning UA as a potential therapeutic for cancer [18]. The mode of delivery of the compound by nano-formulations such as nanoliposomes has not shown much improvement in increased plasma levels [56]. In addition, the Phase I study of nanoliposome formulation showed no cumulative accumulation of UA in the body as performed by the multi-phase pharmacokinetics study. These studies suggest that while UA itself has promise as an anticancer therapeutic, structural modification for the UA molecule to enhance its solubility, absorption, low-protein binding, and longer tissue accumulation should significantly enhance its application in cancer patients.

Concerning the versatile properties of UA, we focused on compiling the synthetic analogs of UA, which show potential anticancer properties, and analyzed their structure-activity relationship (SAR). UA (3-(*β*-hydroxy-urs-12-en-28-oic acid), **1**) is a ubiquitous pentacyclic compound that possesses functional groups such as a carboxylic group at C28, *β*-hydroxy function at C3, and an alkene at C12–C13 (Figure 2). Considering UA as a lead molecule, researchers extensively modified and hybridized UA at these sites with the aim of enhancing its anticancer properties and overcoming the associated poor absorption and low bioavailability drawbacks. We have categorized our analysis based on modifications and/or alterations at different pharmacophoric sites.

## 3. Materials and Methods

According to the databases of PubMed, Science Detect, Web of Science, American Chemical Society, Springer, and Scopus, a comprehensive and systematic review was performed. The keywords “ursolic acid” AND “anticancer” were used and further filtered by “analogs”, “derivatives” and “synthesis”. The systematic search of electronic databases identified 183 articles after excluding patents, clinical trials, and conference abstracts. We carefully reviewed all the articles and chose the articles that discuss and/or report on modified ursolic acids and their anticancer properties.

## 4. Synthetic Analogs of UA

Molecular hybridization (the conjugation of two or more bioactive scaffolds via a covalent bond, MH) is a powerful strategy in drug discovery and can play an effective and efficient tool for the development of new drug candidates. MH is a strategy of rational design of new ligands or prototypes based on the combination of pharmacophoric moieties and has emerged as an important strategy for the development of new hybrid architectures that maintain pre-selected characteristics of the original templates and can act as multitarget ligands. This approach could contribute to the design of multifunctional cancer drug candidates and might be useful in overcoming the problems associated with parent drugs [57,58,59,60,61,62,63,64,65]. In the last two decades, the MH approach has been extensively incorporated in UA research and has reported several modified UA products as anticancer agents [66]. We believe the compiled structure analysis and rationale could help the medicinal chemist to develop potential anticancer drug candidates.

### 4.1. Modification of UA at C3

Several attempts were made to investigate the anticancer effect of the alcohol group at the C3 position of ursolic acid. Thien et al. synthesized compound **2** starting from UA. Compound **2** (Figure 3) was found to be 2–3 times more active than the parent UA against human mouth epidermal carcinoma (KB), human hepatocellular carcinoma (HepG2), human breast carcinoma (MCF7), and human lung carcinoma (Lu) cell lines [67].

Synergistic or combination therapy is one of the promising approaches in cancer treatment. da Silva et al. converted the C3 alcohol group to an amino group (**3**) and used it in combination with imatinib to create the impact of a co-drug on leukemia cells (K562). Compound **3** (Figure 4), itself, shows potential inhibition against K562 (IC_50_ value of 5.2 μM). Further investigation suggests that it induced cell death (apoptosis) via activating caspases-3 and -8 and caused cell cycle arrest. K562 cells treated with Compound 3 were arrested in the G1 phase of the cell cycle and decreased the cell population in the G2 phase. A synergistic effect was observed when **3** was used along with imatinib to treat leukemia [68].

Xu et al. introduced the 3,4,5-methoxy benzoic acid moiety at the C3 position of UA and evaluated their cytotoxicity properties against A549, MCF7, H1975, and BGC823. The UA derivative containing 3,4,5-methoxy-phenacyl at the C3 position (**4**) (Figure 5) shows a significant selective antiproliferative effect with IC_50_ values in the range of 6.07 to 22.27 µM [69].

Another attempt was made to modify the C3 alcohol group of UA at a carbonyl group (**5**) (Figure 6) and also introduce three carbons using the Barbier–Grignard method. The synthesized stereoisomeric analogs of UA (**6** and **7**) were found to be effective against three cancer cell lines (HepG2, Hep3B, and HA22T/VGH). Further studies confirm the inhibition of NF-κB activation. However, analog 5 was ineffective against all three cell lines [70].

UA analogs (modification at C3) listed in Table 1 reduced cell cycle arrest and induced apoptosis by activating Bax and caspases-8, -9, and -3 and also by reducing Bcl2 and MDM2. Apoptosis is involved in maintaining homeostasis of cell numbers by removing damaged or unwanted cells; therefore, UA derivatives could be important chemotherapeutic compounds to reduce tumor growth. UA-derivatives also inhibit NF−κB, which in turn, reduces cell growth arrest. NF−κB signaling plays an important role in cell growth and inhibition. NF−κB signaling will reduce cell growth by inducing cell cycle arrest (Table 1).

### 4.2. Modification of UA at C28

The methyl ester derivative of UA (**8**) (Figure 7) did not affect the anticancer properties of UA, as revealed by several reports [71]. However, the benzyl ester of UA (**9**) showed a decrease in anticancer properties against the K562 leukemia cell line [72]. Nevertheless, the isopropyl ester moiety at C17-COOH (C28) showed a potent inhibitory effect on the growth of the human bladder cancer (NTUB1) cell line. Flow cytometric analysis demonstrated that treatment of NTUB1 with **10** led to cell cycle arrest in addition to an increase in apoptotic cell death. These data suggest that the possibility of G1 phase arrest and apoptosis in **10**-treated NTUB1 for 24 h was mediated through an increased amount of ROS in cells exposed with **10**, while the presence of G2/M arrest before the accumulation of cells in the sub-G1 phase in **7**-treated cells for 48 h was also due to an increased amount of ROS in cells exposed to **10** [73].

Several modifications were made to UA by incorporating a glycosyl or arylpropenoyl scaffold at the C28 and/or C3 positions. The observed experimental data suggest that only compound **11** (Figure 8) was found to have enhanced anticancer properties against human colon adenocarcinoma (HT29) cell lines with an IC_50_ value of 8 µM in comparison to UA (IC_50_: 30 µM) [74].

Bai et al. synthesized a set of UA conjugates with structural modifications at the C3 and C28 positions. Compound **12** (Figure 9) stood out from the pool compound by showing cytotoxicity with hepatoma (HepG2), gastric carcinoma (AGS), colorectal carcinoma (HT29), and prostatic carcinoma (PC3) cell lines. However, compound **12** is more effective against AGS cell lines with an IC_50_ value of 11.4 µM. The cell cycle distribution and sub-G0/G1 ratio data indicate that compound **12** is effective with 92.64 ± 3.13% inhibition. The presence of -OH and -NH_2_ groups contributes to improving the log P. Theoretical (arithmetic) and computer-assistant calculations were then used to predict the logP value, such as ACD\logP and XlogP3 and Molinspiration\logP [75]. The similar analog, **13**, was prepared and tested against MCF7, Hela, and A549 cell lines. Analog **13** show improved IC_50_ values (9.19 ± 0.82, 8.56 ± 0.53, and 12.72 ± 0.79 µM for MCF7, Hela, and A549 cell lines, respectively). In addition, it is also observed to increase its carbon chain length between NH and NH_2_, which decreases the potency [76].

Liu et al. investigated the application of C28 modified UA and found that keeping the C3 alcohol group unchanged as a polar substituent at the C3 position is essential for pharmacological activities. From the series of synthesized conjugates, compound **14** was identified as the most potent against human gastric cancer (MGC803) and human breast cancer (Bcap37) cell lines with an IC_50_ of 2.50 ± 0.25 µM and 9.24 ± 0.53 µM, respectively. Further mechanistic studies, such as acridine orange/ethidium bromide staining, Hoechst 33,258 staining, terminal deoxynucleotidyl transferase biotin-dUTP nick end-labeling (TUNEL) assay, and flow cytometry, indicate **14** (Figure 10) can induce apoptosis in MGC803 cells [77].

The same research group synthesized another series of C28-modified UA based on the above results. Compound **15** was found to be the most active from the new series among all the synthesized analogs. Several variations have been explored with different chain lengths and secondary heterocycles. The combination of our carbon chain linker and piperidine gave the best outcome against MGC803 and Bcap37 human cancer cell lines with 4.53 and >20 µM IC_50_ values, respectively. Further study of the mechanism via acridine orange/ethidium bromide staining, Hoechst 33,258 staining, and TUNEL assay and flow cytometry suggest **15** can induce apoptosis in MGC803 cells at 10 μM [78].

Hypoxia-inducible factor-1α (HIF-1α) is one of the key mediators of angiogenesis and survival in tumor metastasis that has been established as an important cancer drug target. Several UA derivatives incorporating oxadiazole, triazolone, and piperazine moieties at the C28 position were designed and synthesized as HIF-1α inhibitors. Most of the derivatives demonstrated the ability to inhibit the expression of HIF-1α, but **16** (Figure 11) exceptionally inhibited HIF-1α transcriptional activity under hypoxic conditions with IC_50_ = 36.9 µM. None of the UA derivatives showed cytotoxic activity (IC_50_ > 100 µM). These compounds require further investigation [79].

Patnaik et al. synthesized UA hybrids by modification at C28 and the incorporation of zidopropyl-3*β*-hydroxy-urs-12-en-28-oate via a cycloaddition reaction. The synthesized hybrid compounds were tested for their anticancer potential against two human breast cancer cell lines (MCF7 & MDA-MB231) using a sulfarhodamine B cell proliferation assay. These hybrids showed more selectivity towards the MDA-MB231 cell line than MCF7. Compound **17** (Figure 12) is the highly effective one, with a GI_50_ value of 1.4 ± 0.1 μM, which is 2.9 times more active than the standard doxorubicin against MDA-MB231. In addition, a mechanistic study suggested that compound **17** arrests cells in the mitotic phase of the cell cycle [80].

Semenova et al. synthesized a set of 1,2,4 triazole-incorporated UA derivatives. The conjugates were evaluated for their antiproliferative activity on MCF7, U-87 MG (glioblastoma multiform cells), A549 (lung carcinoma), and HepG2 (hepatocarcinoma) cell lines. The compounds were moderately active and, interestingly, they showed selectivity towards A549. Compounds **18** and **19** (Figure 13) are azole-based thiones that show selective inhibition (IC_50_ values 11.25 ± 0.77 and 13.45 ± 1.14 µM) [81].

UA conjugated with a lipophilic triphenylphosphonium cation was screened with three human cancer cell lines (MCF7, HCT116, and TET21N). The triphenylphosphonium UA derivative (**20**) (Figure 14) not only showed higher cytotoxicity as compared to UA but was also markedly superior in triggering mitochondria-dependent apoptosis, as assessed using a range of apoptosis markers such as cytochrome c release, stimulation of caspase-3 activity, and cleavage of poly(ADP-ribose) polymerase [82].

The published studies indicate that these C28 modified UA derivatives induce apoptotic cell death by activating death-caspases, caspase-3 or/and caspase-8, unlike the C3 derivatives, which seem to have caused cell cycle arrest and mitotic catastrophe-induced cell death. However, a lack of data on non-proliferating cells or cells with intact p53 makes these studies hard to compare for their tumor cell specific toxicity. Further, a lack of testing in vivo is another weakness when assessing their potential as future antitumor drugs (Table 2).

### 4.3. Modification of UA at Both C3 and C28

Esterification of 3-OH and 17-COOH significantly reduces the anticancer properties [71,72]. Maintaining a polar group at either position is essential for the activity [83]. Popov and co-workers explored the variation on UA at C3 and C28 by introducing 1,3.4-oxadiazole, 1,3.4-oxadiazole, and/or 1,2,3-triazole moieties. All the possible analogs were synthesized and screened against different cancer cell lines along with immortalized human fibroblasts. Hybrid conjugates of 1,3,4-oxadiazoles attached at the C3 and C28 positions of the UA frame via a triazole spacer, and combinations of 1,2,5-oxadiazole or 1,3,4-oxadiazole, tethered with a succinate linker, and 1,2,3-triazole at C3 of the UA backbone, were found inactive. From the series of conjugates, **22a** and **22b** (Figure 15) show the best cytotoxic activity and selectivity on HepG2 and MCF7 cells. The introduction of an additional ester-type linker between triazole (**24**) and the UA scaffold or UA frame and furoxan, either at C3 (**23**) or at C28 (**25**), led to the loss of cytotoxicity (Table 3) [84].

Shao et al. developed several UA analogs, retaining the polar properties at the C3 and C28 positions. Among the several synthesized analogs, compound **26** (Figure 16) showed potential anticancer properties against human hepatoma (HepG2), human gastric cancer (BGC823), human neuroblastoma (SH-SY5Y), human cervical carcinoma (Hela), and human embryonic lung fibroblast (HELF) cell lines. Further, a flow cytometric analysis and morphologic changes study suggested compound **26** arrests cell cycle progression at the S phase in HepG2 cells [85]. The same compound **26** was prepared by another research group, who evaluated its EC_50_ values against several cancer cell lines; they validated the potency of **26**, however, it also had a lower ED_50_ against the non-malignant fibroblasts NIH 3T3 [86].

UA-derived hydroxamates were synthesized by adding a C3 acetyl group; they were screened for their cytotoxicity utilizing SRB assays against several human tumor cell lines. Among the various hydroxamates, compound **27** (Figure 17) was found to be the most potent against seven cancer cell lines. However, unfortunately, these hydroxamates were nonselective and showed toxicity to nonmalignant mouse fibroblasts (NIH 3T3) [87].

The structure–activity relationship confirms the acetyl group at C3 is more effective than the butyl group. Keeping the acetyl group at the C3 position, another set of conjugates was developed with variations at the C28 position. Compound **28** (Figure 18) was found to be more potent than the other synthesized ones against Hela, SKOV3, and BGC823 cell lines [88].

Another similar modification was attempted to prepare compound **29** (Figure 19), which targeted gastric cancer. Compound **29** was tested against BGC823 cells and a human normal gastric (GES1) cell line to demonstrate its selectivity. The studies uncovered that **29** could induce apoptotic effects in the treated BGC823 cells, such as a comet-like DNA bend, sub-G0/G1 phase accumulation, and phosphatidylserine externalization (apoptosis). In addition, the activity of caspase-3 was found to be up-regulated, while the expression of Bcl2 and survivin were down-regulated in **29** treated cells. Data from studies in mice indicate compound **29** is safer than Taxol [89].

Compound **30** was synthesized to have an acetyl group at the C3 position. The modified UA **30** was 2–3 times more active than the parent UA against human mouth epidermal carcinoma (KB), human hepatocellular carcinoma (HepG2), human breast carcinoma (MCF7), and human lung carcinoma (Lu) cell lines [67].

It is important to understand the impact of harsh environments on cancer cells as several discoveries on the metabolic pathways of cancer cells have led to a better overall understanding of their ability to proliferate and adapt to their microenvironment [90]. To uncover the pathways involved in cancer cells, or highly proliferative tumors, recently “cancer metabolism” and “metabolic reprogramming” have been investigated [91,92,93,94,95]. A new analog (**31**) (Figure 20) was designed by using a computer-aided drug design (CADD) approach to have an acetyl group at C3. The molecular docking studies suggest **31** could bind to the active sites of glucokinase (GK), glucose transporter 1 (GLUT1), and ATPase, which are the key enzymes involved in cancer glucose metabolism. Further, experimental observations confirm the synergistic effect of **31** and glycolysis inhibitor 2-deoxy-D-glucose (2-DG) in inhibiting the glucose metabolism of cancer cells. The depletion of intracellular ATP and decrease in lactate production triggers the cancer cell’s arrest in the S and G2/M cycle phases. In addition, the combination selectively down-regulated the expression of the Bcl2 and HKII proteins and up-regulated the expression of Bax and p53, which results in enhanced apoptosis. The Western blotting assay illustrates the molecular targets of **31**, which includes Bax, Bcl2, p53, and HK proteins. The LC-MS analysis of animal serum proved the bioavailability of **31** [96].

Yang et al. synthesized a similar set of UA analogs containing the C3 acetyl group. The synthesized UA analogs were inhibiting cell growth when assayed against various tumor cell lines and a non-pathogenic cell line of normal human embryonic lung fibroblasts (HELF). A theoretical toxicity risk analysis was carried out using OSIRIS and the results indicated that most compounds showed moderate to low risks. However, compound **32** stands alone with significant IC_50_ values ranging from 4.09 ± 0.27 to 7.78 ± 0.43 µM against 12 different tumor cell lines. Flow cytometry analysis data indicated that **32** induced G0/G1 arrest in three of these cell lines (HT29, HepG2, and RL95-2). Observed experimental data were validated by structural docking studies, which confirmed that UA or its derivatives, could bind to cyclins D1 (CycD1) and cyclin-dependent kinases (CDK6), the key regulators of G0/G1 transition in the cell cycle, while the piperazine moiety of **32** could bind with glucokinase (GK), glucose transporter 1 (GLUT1), and ATPase, which are the key proteins involved in cancer cell metabolism. Further, the inducing apoptosis capability of **32** was confirmed by acridine orange/ethidium bromide staining and decreased cell viability in a dose-dependent mode [97].

Keeping the acetyl group at the C3 position intact, the C28 position was further explored, and several analogs were synthesized and screened against five cancer cell lines (MGC803, HCT116, T24, HepG2, and A549 cell lines) and a normal cell (HL7702) using an MTT assay. Compound **33** was identified as the most potent analog and investigated via cell distribution by acridine orange/ethidium bromide staining, Hoechst 33,258 staining, JC-1 mitochondrial membrane potential staining, and flow cytometry, which confirmed that the potency of **33** (Figure 21) was probably achieved via the induction of cell apoptosis by G1 cell-cycle arrest. Further, Western blot and qRT-PCR (quantitative real-time PCR) experiments confirmed that **33** induces apoptosis via both intrinsic and extrinsic mechanisms [98].

Another set of piperazine-like UA conjugates were developed and tested with several cancer cell lines. Among the conjugates, **34** and **35** (Figure 22)were shown to be potent against several cancer cells, however, these showed high toxicity on nonmalignant fibroblasts [99].

Transcription factor nuclear factor-kappa B (NF-κB), known to be a regulator of a wide variety of anti-apoptotic proteins, is commonly over-expressed and constitutively activated in different types of hematologic cancers and solid tumors. NF-κB is one of the prime targets for cancer therapy. A series of UA analogs containing long-chain diamine moieties were designed as NF-κB inhibitors. Compound **36** shows anticancer potential against the test tumor cell lines including multidrug-resistant human cancer lines, with the IC_50_ values ranging from 5.22 to 8.95 μM. A further mechanistic study revealed compound **36** (Figure 23) arrests the cell cycle at the G1 phase and triggered apoptosis in A549 cells through blockage of the NF-κB signaling pathway [100].

A set of ethylenediamine-spacer carboxamides of UA (**37** and **38**) (Figure 24) were prepared and evaluated for their cytotoxicity against several human tumor cell lines using SRB assays. All the synthesized carboxamides showed increased cytotoxicity for the cancer cells than UA. However, unfortunately, the toxicity was not selective and exerted enhanced toxicity against non-malignant mouse fibroblasts (NIH 3T3). The presence of the acetyl group in these carboxamides helps in increasing the cytotoxicity. These carboxamides require further attention to improve their selectivity [101].

Polyamines contain positively charged nitrogen atoms at physiological pH and can serve as electrostatic bridges between negatively charged phosphates of DNA. Polyamines are precursors of aminoalkylguanidines, which could be considered as a scaffold for the development of chemotherapeutic agents [102,103]. Spivak et al. synthesized a C28 Guanidine-Functionalized UA analog (**39**) (Figure 25). The impact of the guanidine group on the antitumor properties of UA was examined and **39** showed enhanced selective cytotoxicity with five human cancer cell lines (Jurkat, K562, U937, HEK, and Hela). Mechanistic study indicated the cell cycle was arrested at the S-phase of Jurkat cells and led to apoptosis [104].

A library of UA derivatives was prepared by incorporating 2-mercapto-1,3,4-oxadiazoles, 2-amino-1,3,4-oxadiazoles, and 3-mercapto-1,2,4-triazoles at the C28 position. From the library, the compound having 2-amino-1,3,4-oxadiazole scaffold (**40**) (Figure 26) was found to be selective towards MCF7 cell lines and showed an anticancer potential comparable to doxorubicin. In addition, the acetyl group at the C3 position enhanced activity in comparison to the deacetylated analog (**41**) [105].

Ma et al. synthesized several analogs of UA via structural modification at C3, C28, and C11 and investigated their anticancer properties. Among all the modifications, compound **42** (Figure 27) was identified as the most potent against human leukemia cancer (HL-60), human gastric cancer (BGC), human hepatocellular carcinoma (Bel7402), and human cervical cancer (Hela) cell lines. The chirality also played an important role as the *β*-form (**42a**) is 20-fold more active than the *α*-form (**42b**) as well as UA [71].

The incorporation of a triazole moiety is one of the most efficient and most adopted approaches in developing drug candidates [106,107]. Rashid et al. synthesized several ursolic acid-triazolyl derivatives and screened them against A549 (lung), MCF7 (breast), HCT116 (colon), THP1 (leukemia), and a normal human epithelial cell line (FR2) using the sulforhodamine-B assay. Four compounds (**43a**–**d**) (Figure 28) out of eighteen compounds were found to be most active. Compounds **43a**–**c** with *o*-bromo, *o*-methoxy and *o*-chloro substitution at the aromatic ring were selective towards cancer cells, however, **43a** showed cytotoxicity for normal cell line (FR-2) cells (Table 4) [108].

UA derivatives were prepared with a modification at the positions C3 and C28 of UA. Compounds **44a** (Figure 29) and **44b** were found to be most active against Hela, HepG2, and BGC823 cell lines in comparison to the reference drug, gefitinib (IC_50_ value 17.1, 20.7, and 19.3 µM for Hela, HepG2, and BGC-823, respectively) [109].

Meng et al. synthesized eighteen UA derivatives with C3 and C28 modifications. Among them, compound **45** (Figure 30) was identified as the most active against BEL7402 and SGC7901 cell lines. The carbon chain length impacts the potency and from the analogs, it indicates a six-carbon chain is optimal [110].

Zhao et al. synthesized UA analog (**46**) (Figure 31) by modifying at the C3 and C28 positions and incorporating an O-[4-(1-piperazinyl)-4-oxo-butyryl moiety. Compound **46** showed 8–10-fold enhanced anticancer properties (against MCF7, Hela, and A549 cell lines) in comparison to the parent UA [111].

Kahnt et al. reported several UA-1,4,7,10-tetraazacyclododecane-1,4,7,10-tetraacetic acid (DOTA) conjugates with various oligo-methylene diamines as linkers. From them, compound **47** (Figure 32) was found to be the most promising and showed EC_50_ values of 1.5 µM (for A375 melanoma) and 1.7 µM (for A2780 ovarian carcinoma) [112].

The hypoxia-inducible factor-1α (HIF-1α) pathway has been implicated in tumor angiogenesis, growth, and metastasis. In the continuous effort to develop potential chemotherapeutics, another potent HIF-1α inhibitor (**48**) (Figure 33) was reported by Zhang et al. with an IC_50_ value 0.8 ± 0.2 µM. The tetrazole located at C28 is the critical component for the potency [113].

UA analogs generated upon modification of both the C3 and C28 positions, reduced cell cycle arrest and induced apoptosis by activating p53, Bax and caspases-8, -9, and -3, and also by reducing Bcl2 and survivin. Apoptosis is involved in maintaining homeostasis of cell numbers by removing damaged or unwanted cells; therefore, UA derivatives are important chemotherapeutic compounds to reduce tumor growth. UA derivatives also inhibit NF−κB, which in turn, reduces cell growth arrest. NF−κB signaling plays an important role in cell growth and the inhibition of NF−κB signaling will reduce cell growth by inducing cell cycle arrest (Table 5).

### 4.4. Modification of UA at Other Positions

In addition to C3 and C28 modifications, several structural changes are reported that help enhance anticancer properties. Leal et al. synthesized a series of heterocyclic derivatives of UA as therapeutics for pancreatic cancer. Compound **49** (Figure 34), which is an α,β-unsaturated ketone at C3 in conjugation with a heterocyclic ring at C28, shows 7-fold more antiproliferative activities than UA with an IC_50_ of 1.9 µM against the pancreatic cancer (AsPC1) cell line. Further investigation confirms **49** arrests the cell cycle in the G1 phase and induces apoptosis in AsPC1 cells with the up-regulation of p53, p21^waf1^ and NOXA protein levels [114]. The same research group developed another potent compound, **50**, which is showing anticancer properties against various cancer cell lines at significantly low concentrations. Compound **50** also arrests the cell cycle at the G1 phase with the up-regulation of p21waf1. Apoptosis was induced at an inhibitor concentration of 8 µm with up-regulation of NOXA and down-regulation of c-FLIP [115].

UA analogs were prepared using a Claisen Schmidt condensation reaction with aromatic aldehydes with the aim of developing potential anticancer drug candidates. Compound **51** (Figure 35) was found to be selective and has shown potency against four human carcinoma cell lines, including A549 (lung), MCF7 (breast), HCT116 (colon), and THP1 (leukemia), at significantly low concentrations. Further mechanistic studies concluded that **51** induced apoptosis in HCT-116 cell lines through the mitochondrial pathway, arrested the cell cycle in the G1 phase, caused accumulation of cytochrome c in the cytosol, and increased the expression levels of the caspase-9 and caspase-3 proteins [116].

In an attempt to develop potent anticancer agents, a set of carbazole derivatives of UA were prepared. Among the various carbazoles, compound **52** (Figure 36) showed promising IC_50_ values of 1.08 ± 0.22 and 1.26 ± 0.17 µM against hepatocarcinoma cell lines (SMMC7721 and HepG2), respectively, which are comparable to doxorubicin. The reduced cytotoxicity against noncancerous LO2 cells (IC_50_ value of 5.75 ± 0.48 lM) makes **52** a potential lead molecule [117].

Similarly, Wang et al. synthesized a set of 2-amino-4-aryl-pyrimidine derivatives of UA and tested their anticancer potential against MCF7 and Hela cells. Compound **53** showed IC_50_ values of 0.48 ± 0.11 and 0.74 ± 0.13 mM for MCF7 and Hela cells, respectively, and significantly low cytotoxicity to LO2 cells. A further molecular mechanistic study revealed that compound **53** inhibits cell migration, induces cell cycle arrest at the S phase, and triggers mitochondrial-related apoptosis by increasing the generation of intracellular ROS and decreasing the mitochondrial membrane potential (MMP). In addition, it participates in the up-regulation of the protein expression level of Bax and the downregulation of the level of Bcl2. Furthermore, molecular docking indicates that MEK1 kinase could be one of the possible targets for **53** [118].

Gu et al. synthesized quinoline derivatives of UA and evaluated their in vitro cytotoxicity against three human cancer cell lines (MDA-MB231, Hela, and SMMC7721). Compound **54** (Figure 37) was found to be the most active one, with IC_50_ values of 0.61 ± 0.07, 0.36 ± 0.05, 12.49 ± 0.08 µM against MDA-MB231, Hela, and SMMC7721 cells, respectively, stronger than the positive control, etoposide. The mechanistic study by the Annexin V-FITC/PI dual staining assay confirmed that compound **54** induces the apoptosis of MDA-MB231 cells in a dose-dependent manner and arrests the cell cycle MDA-MB231 cells at the G0/G1 phase [119].

As we discussed earlier, an aminoguanidine moiety scaffold plays an important role in the development of potential drug candidates. Here, Wu et al. introduced the scaffold at the C3 position of the UA and synthesized compound **55** (Figure 38). The compound was tested for the inhibition of HIF-1α transcriptional activity under hypoxia conditions using a Hep3B cell-based luciferase reporter assay (IC_50_ = 4.0 µM). Further investigation indicated the down-regulation of HIF-1α protein expression, possibly by suppressing its synthesis, reducing the production of vascular endothelial growth factor, and inhibiting the proliferation of the cancer cells [120].

Nitric oxide (NO) is an important messenger molecule in the human body that plays a key role in many physiological processes. High levels of NO can inhibit the proliferation of cancer cells, but unfortunately, the in vivo half-life of NO is very short. Recently, NO-donating derivatives, which can transport and release NO within the body, were used to explore the development of chemotherapeutic agents [121,122]. The approach was adopted, and a set of NO-donor UA derivatives were prepared and evaluated for their in vitro cytotoxicity against four human cancer cell lines (HepG2, MCF7, HT29, and A549). Compound **56** was found to be the most active with IC_50_ 4.28 μM against HT29. An additional bioassay explained that this compound induced cell cycle arrest at the G1 phase and apoptosis in a dose-dependent manner. In addition, compound **56** (Figure 39) was found to up-regulate pro-apoptotic Bax, p53, and down-regulate antiapoptotic Bcl2 [123]. Another potential NO-donating UA–benzylidene derivative (**57**) was developed, which showed a 3- to 9-fold higher cytotoxicity as compared with the parent drug in A549, MCF7, HepG2, HT29, and HeLa cells. Further analysis concluded that **57** arrested the MCF7 cell cycle in the G1 phase, which was associated with caspase activation and a decrease in the Bcl2/Bax ratio. A molecular docking study unveiled that the nitroxyethyl moiety of **57** possibly establishes hydrogen bonds with caspase-8 amino acid residues (SER256 and HIS255) [124].

A series of indolequinone derivatives of UA having ester, hydrazide, or amide moieties were prepared and tested for their in vitro antiproliferative properties against MCF-7, Hela, and HepG2 cell lines and a normal gastric mucosal cell line (Ges1). From the series, compound **58** (Figure 40) was identified as the most potent, with IC_50_ values of 1.66 ± 0.21, 3.16 ± 0.24, and 10.35 ± 1.63 μM, respectively, against the cancer cell lines. Detailed molecular mechanism studies disclosed that compound **58** could arrest the cell cycle at the S phase, suppress the migration of MCF-7 cells, elevate the intracellular reactive oxygen species (ROS) level, and decrease mitochondrial membrane potential. In addition, the Western blot analysis confirmed that compound **58** up-regulated Bax, cleaved caspases-3 and -9, cleaved PARP levels, and down-regulated the Bcl2 level of MCF7 cells. Meanwhile, compound **58** markedly decreased p-AKT and p-mTOR expression, which revealed that compound **58** probably exerts its cytotoxicity by targeting the PI3K/AKT/mTOR signaling pathway [125].

Secondary amines are the key building blocks of many pharmaceutically active compounds, such as gefitinib and imatinib [126,127]. Wang et al. introduced the piperazine and morpholine rings at the C3 position of a modified UA via a linker. The piperazine-containing analog **59** (Figure 41) was identified as being more potent against three different cancer cell lines than the morpholine-containing one, **60** [128].

Other analogs of UA listed in Table 6 reduced cell cycle arrest by inducing p21 and induced apoptosis by activating Bax, Noxo, and caspases-8, -9, and -3 and also by reducing Bcl2 and cFLIP. Apoptosis is involved in maintaining homeostasis of cell numbers by removing damaged or unwanted cells and, therefore, UA derivatives are important chemotherapeutic compounds for reducing tumor growth. UA derivatives also inhibit RAS/RAF/ERK/MAPK signaling pathways, which are involved in cell growth promotion.

## 5. Conclusions and Future

Structural modification of some complex natural products, such as UA, offer an unrivaled opportunity to develop novel anticancer compounds with diverse biological activities and limited systemic toxicity. As the description of the various classes of UA modified to enhance their anti-proliferative activities show, tremendous potential for modifications exists both structurally and functionally. The structural modifications of UA showed increased potency and diverse targets, able to inhibit processes responsible for cancer pathology, such as altered energy metabolism, mitochondrial functions, cancer cell migration, and an invasive potential, which are critical for sustaining malignancy and progression.

There are still plenty of opportunities to explore UA and design potential therapeutics for cancer. Fewer reports were found on the C3 modification because there is the assumption that the presence of a polar group at the C3 position is essential for the anticancer properties. Several reports confirm that the acetyl group at the C3 position also increases anticancer properties. In addition, the incorporation of triazole, piperazine, and guanidine scaffolds enhances activity and selectivity. In addition, including an NO donor is one of the interesting approaches.

A recognized deficiency in many of these studies is the lack of data concerning the efficacy of these altered UA analogs in vivo. Although in vitro studies are indicative of the expected functions of the compounds on tumor cells, favorable pharmacological properties, such as pharmacokinetics, and more importantly, pharmacodynamics, in either model animals (usually rodents) or in human volunteers, will move these compounds from the laboratory bench to the clinic. These studies are economically challenging for most laboratories engaged in creating analogs. However, select compounds, such as compounds **33a**–**d**, **40**, and **41** offer significant potential to be successful anticancer compounds in vivo. Perhaps future pharmacological studies for bioavailability, systemic toxicity, and antitumor efficacy may prove some if not all of these compounds have clinical use.

## Figures and Tables

**Figure 1 molecules-27-08981-f001:**
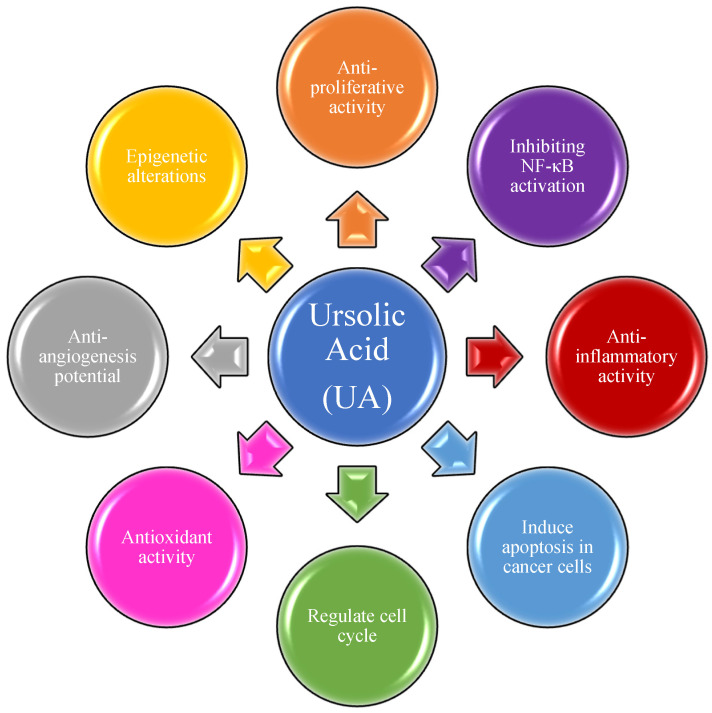
Ursolic acid and its molecular mechanism of action in cancer treatment.

**Figure 2 molecules-27-08981-f002:**
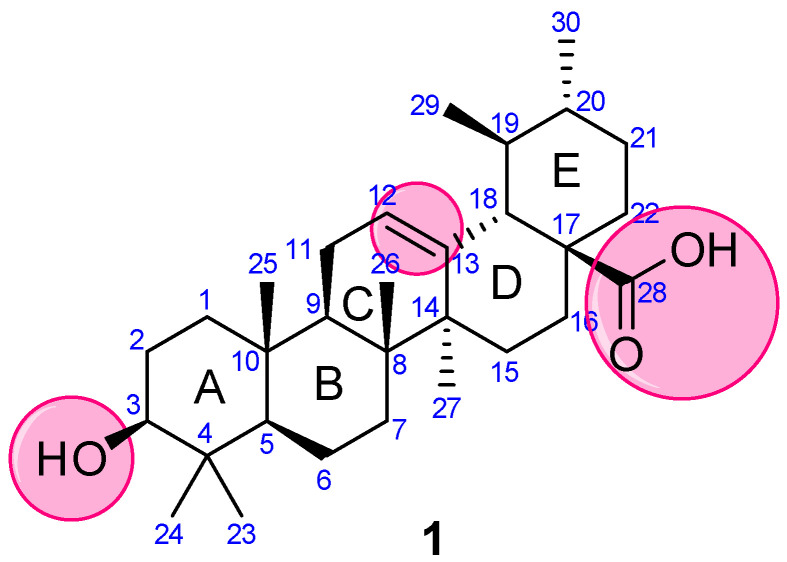
Structure of UA with highlighted pharmacophoric sites.

**Figure 3 molecules-27-08981-f003:**
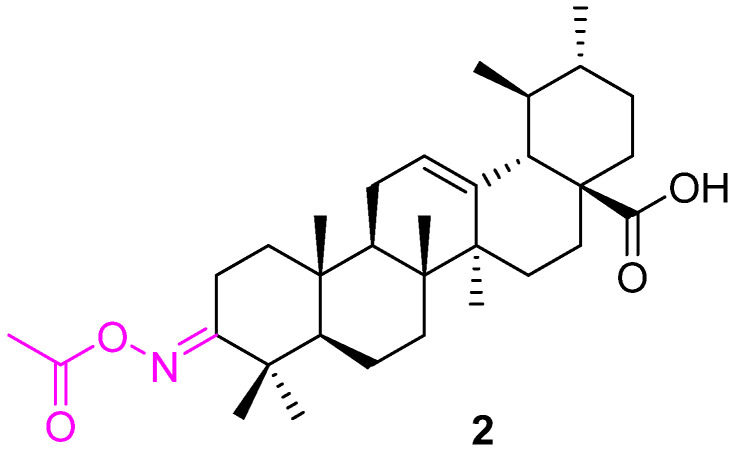
Structure of **2**.

**Figure 4 molecules-27-08981-f004:**
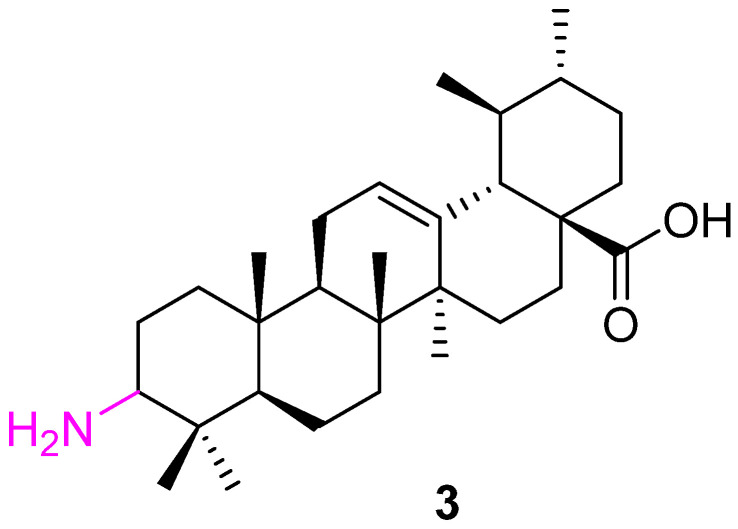
Structure of **3**.

**Figure 5 molecules-27-08981-f005:**
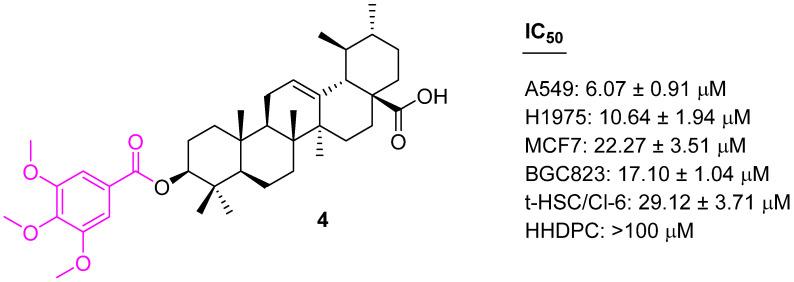
Structure of **4** which shows antiproliferative effect.

**Figure 6 molecules-27-08981-f006:**
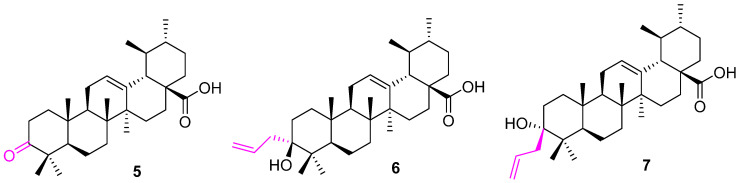
Structure of **5**, **6**, **7**.

**Figure 7 molecules-27-08981-f007:**
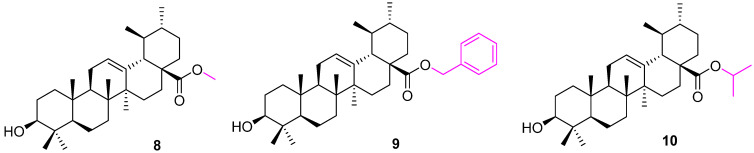
Structure of **8**, **9**, **10**.

**Figure 8 molecules-27-08981-f008:**
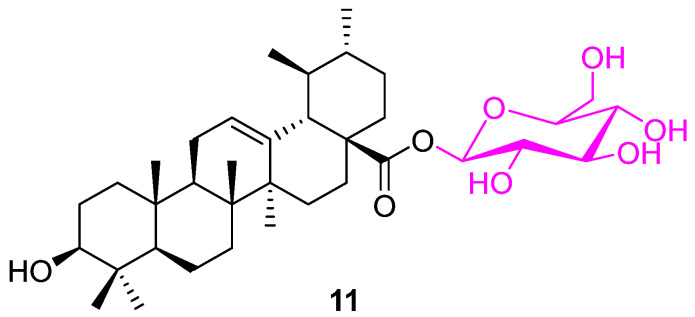
Structure of **11**.

**Figure 9 molecules-27-08981-f009:**
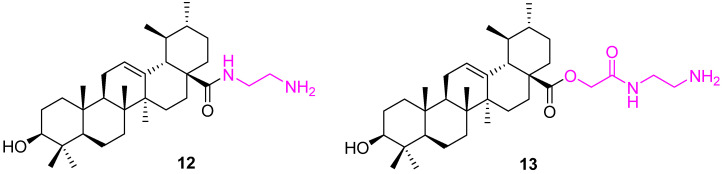
Structure of **12**, **13**.

**Figure 10 molecules-27-08981-f010:**
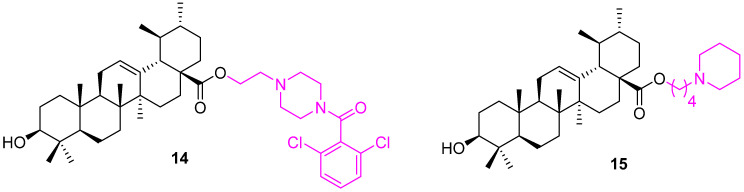
Structure of **14**, **15**.

**Figure 11 molecules-27-08981-f011:**
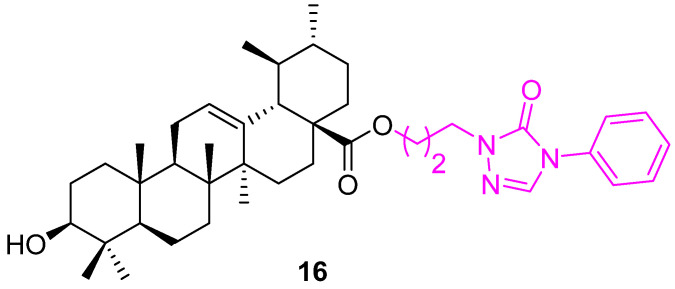
Structure of **16**.

**Figure 12 molecules-27-08981-f012:**
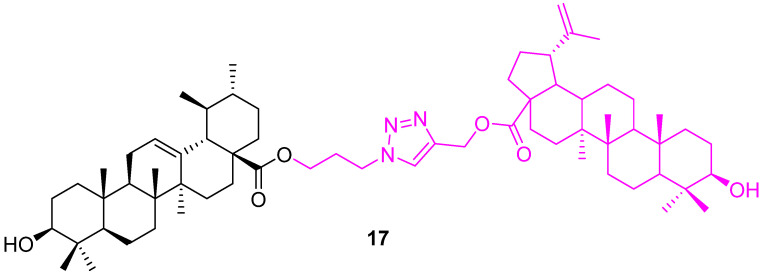
Structure of **17**.

**Figure 13 molecules-27-08981-f013:**
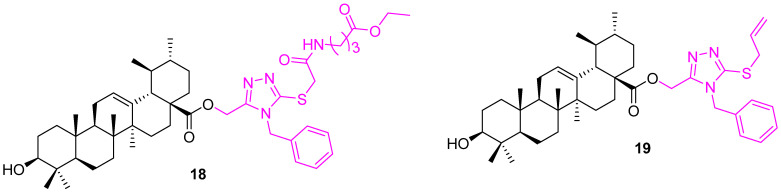
Structure of **18**, **19**.

**Figure 14 molecules-27-08981-f014:**
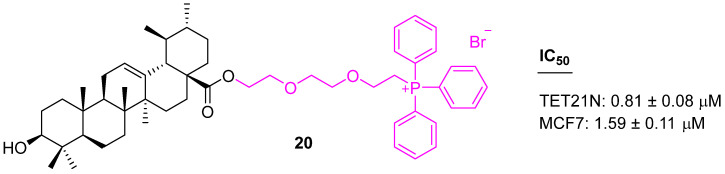
Structure of **20**.

**Figure 15 molecules-27-08981-f015:**
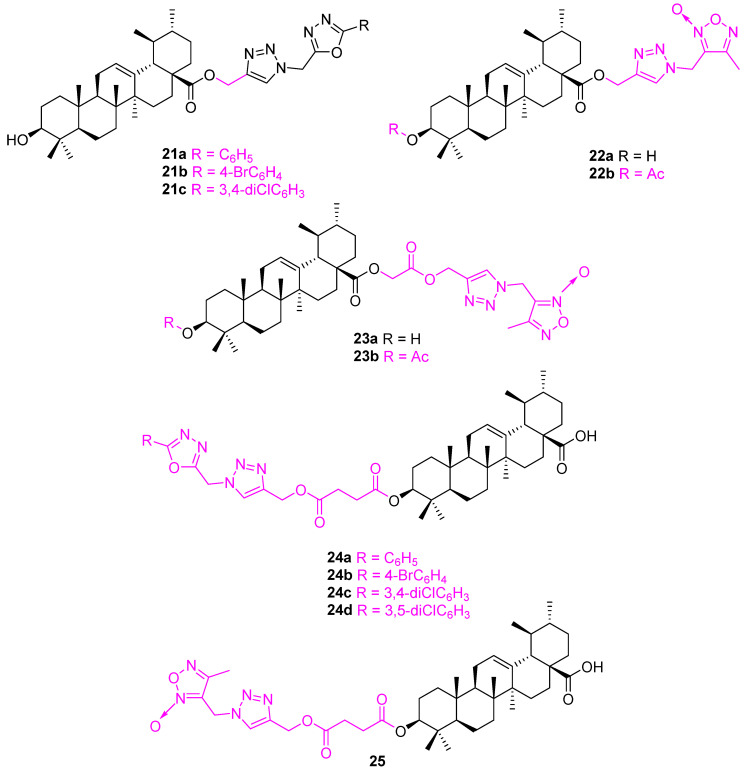
Structure of **21a**–**c**, **22a**, **22b**, **23a**, **23b**, **24a**–**d**, **25**.

**Figure 16 molecules-27-08981-f016:**
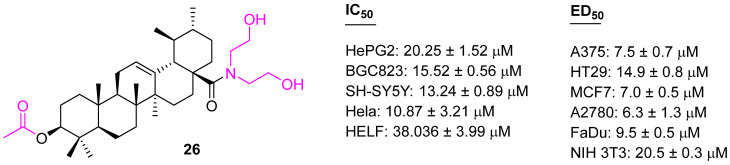
Structure of **26**.

**Figure 17 molecules-27-08981-f017:**
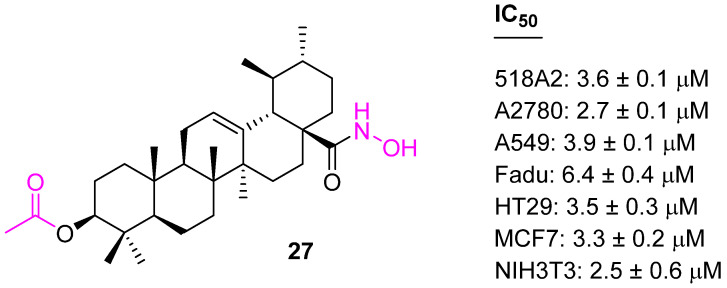
Structure of **27**.

**Figure 18 molecules-27-08981-f018:**
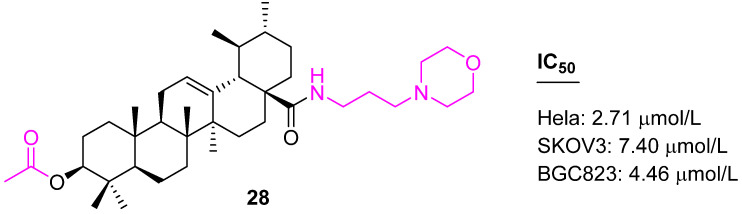
Structure of **28**.

**Figure 19 molecules-27-08981-f019:**
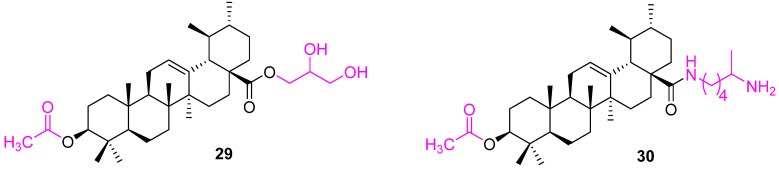
Structure of **29**, **30**.

**Figure 20 molecules-27-08981-f020:**
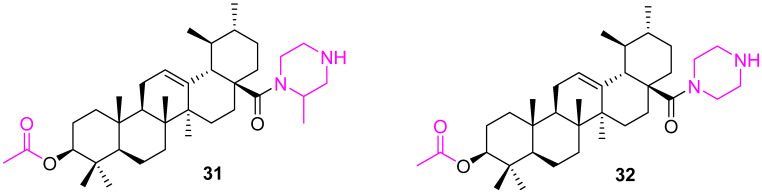
Structure of **31**, **32**.

**Figure 21 molecules-27-08981-f021:**
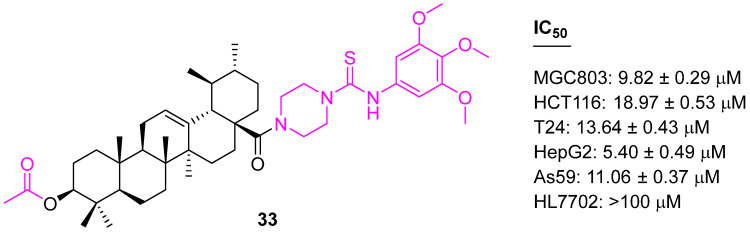
Structure of **33**.

**Figure 22 molecules-27-08981-f022:**
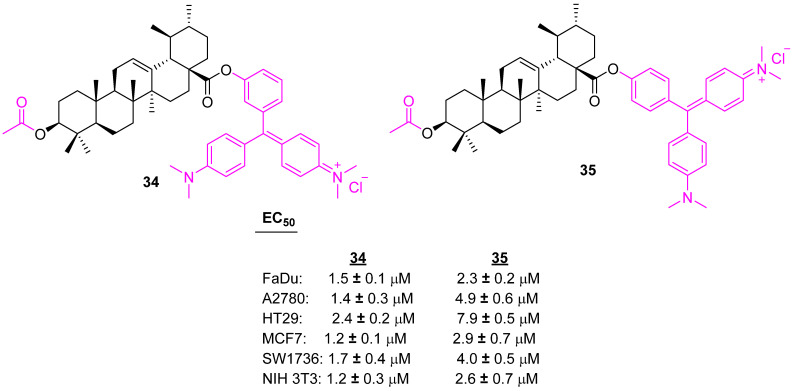
Structure of **34**, **35**.

**Figure 23 molecules-27-08981-f023:**
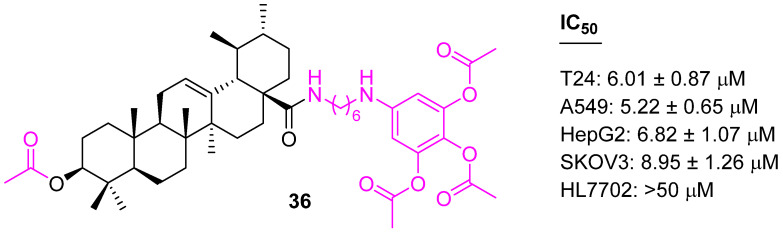
Structure of **36**.

**Figure 24 molecules-27-08981-f024:**
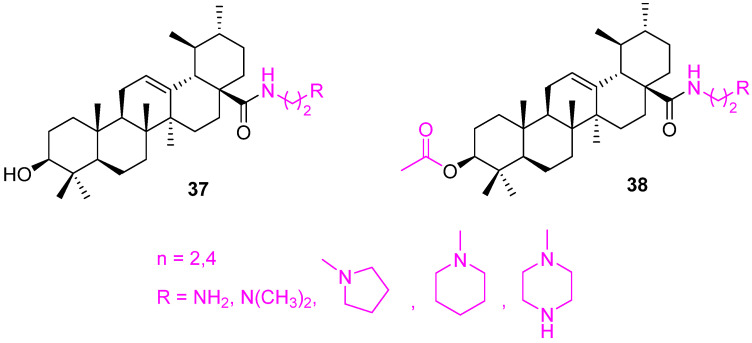
Structure of **37**, **38**.

**Figure 25 molecules-27-08981-f025:**
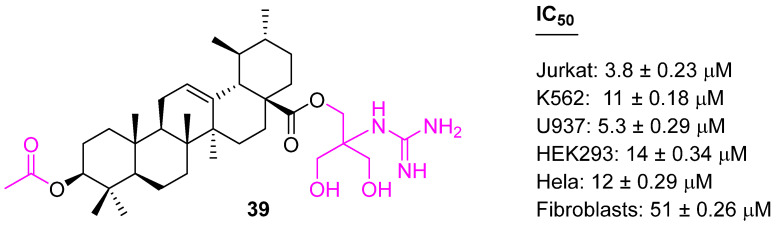
Structure of **39**.

**Figure 26 molecules-27-08981-f026:**
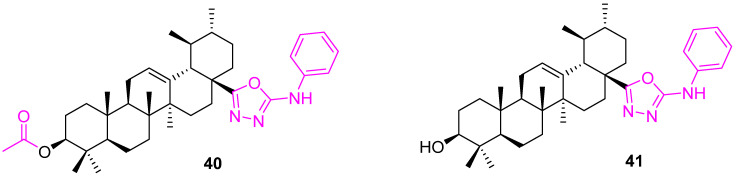
Structure of **40**, **41**.

**Figure 27 molecules-27-08981-f027:**
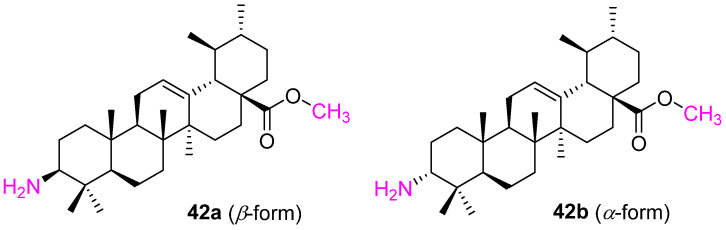
Structure of **42a**, **42b**.

**Figure 28 molecules-27-08981-f028:**
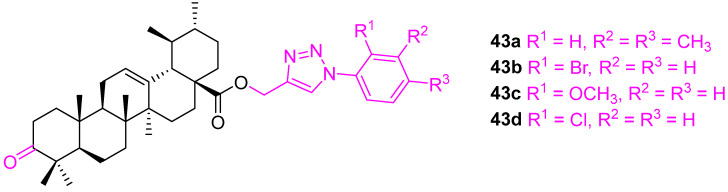
Structure of **43a**–**d**.

**Figure 29 molecules-27-08981-f029:**
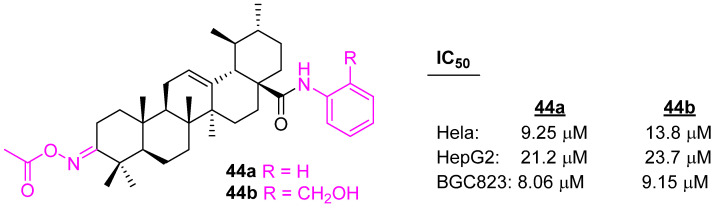
Structure of **44a**, **44b**.

**Figure 30 molecules-27-08981-f030:**
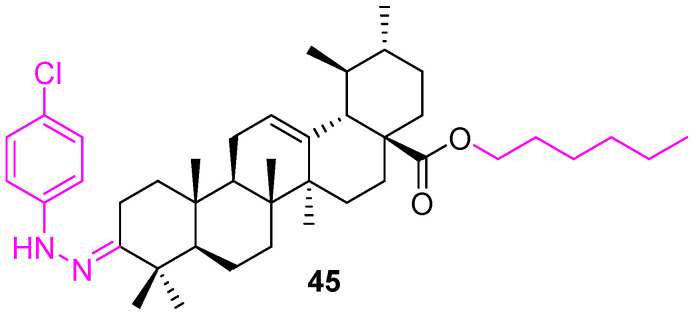
Structure of **45**.

**Figure 31 molecules-27-08981-f031:**
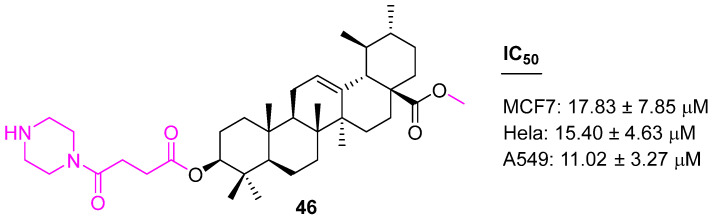
Structure of **46**.

**Figure 32 molecules-27-08981-f032:**
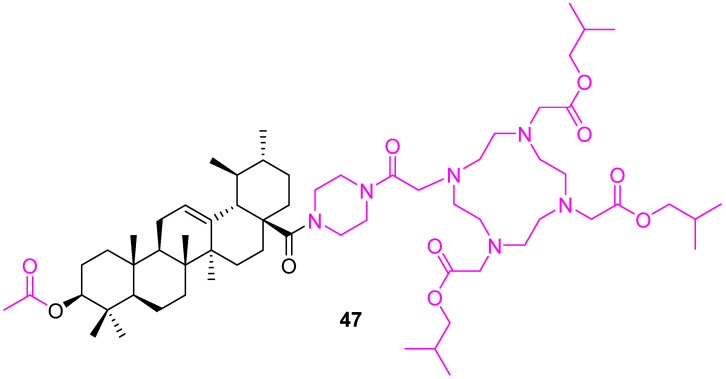
Structure of **47**.

**Figure 33 molecules-27-08981-f033:**
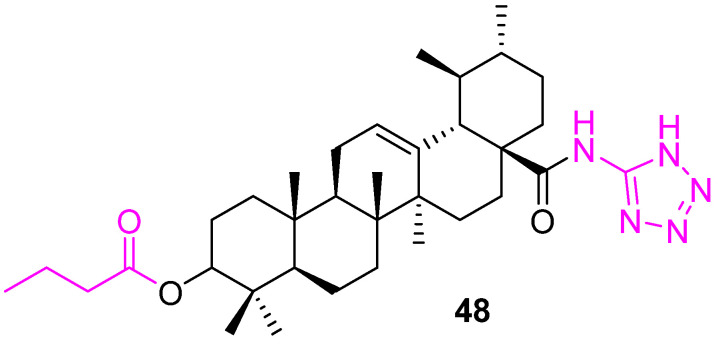
Structure of **48**.

**Figure 34 molecules-27-08981-f034:**
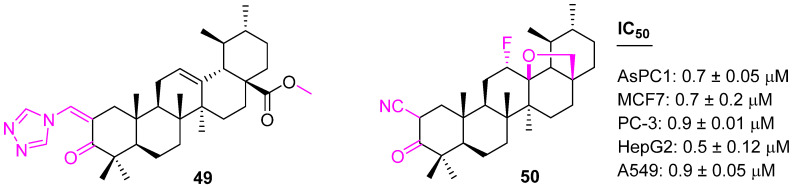
Structure of **49**, **50**.

**Figure 35 molecules-27-08981-f035:**
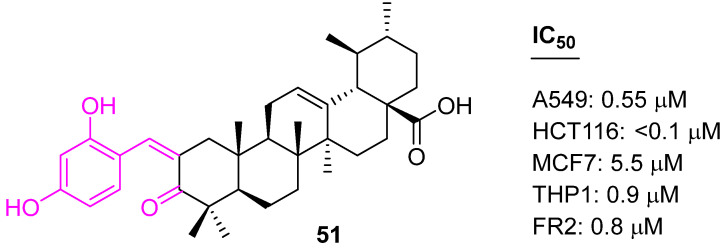
Structure of **51**.

**Figure 36 molecules-27-08981-f036:**
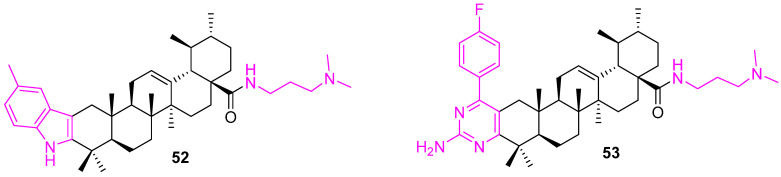
Structure of **52**, **53**.

**Figure 37 molecules-27-08981-f037:**
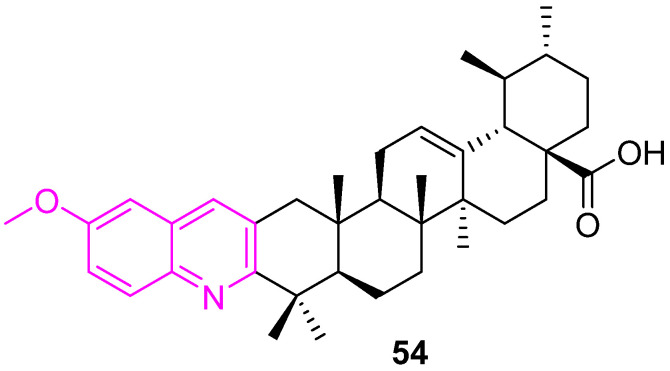
Structure of **54**.

**Figure 38 molecules-27-08981-f038:**
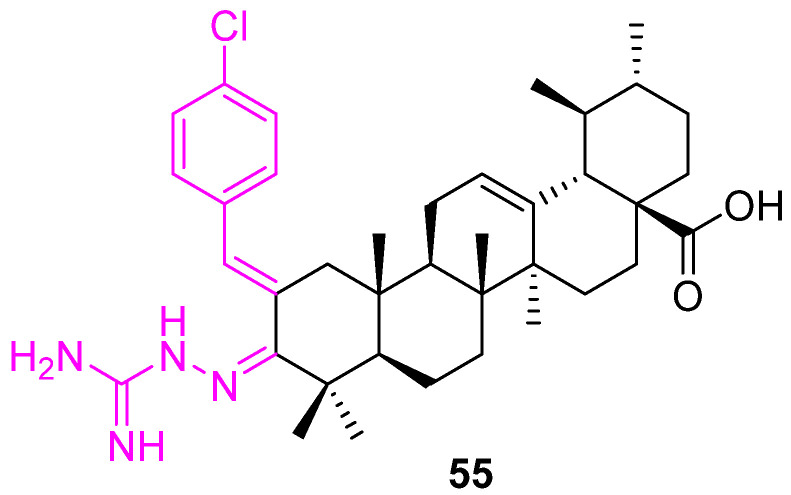
Structure of **55**.

**Figure 39 molecules-27-08981-f039:**
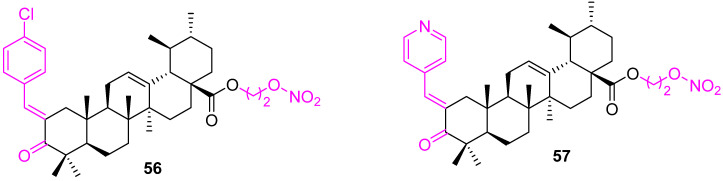
Structure of **56**, **57**.

**Figure 40 molecules-27-08981-f040:**
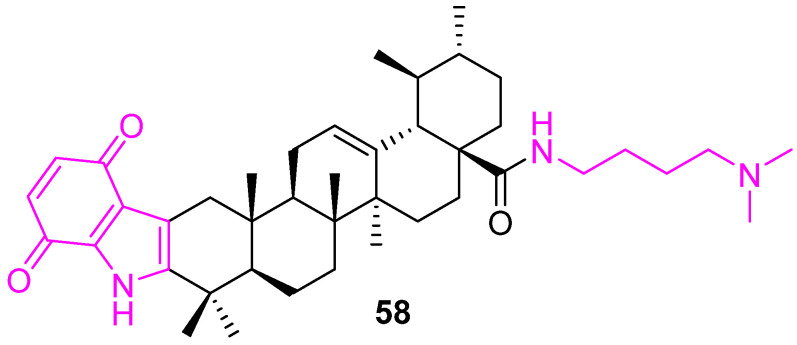
Structure of **58**.

**Figure 41 molecules-27-08981-f041:**
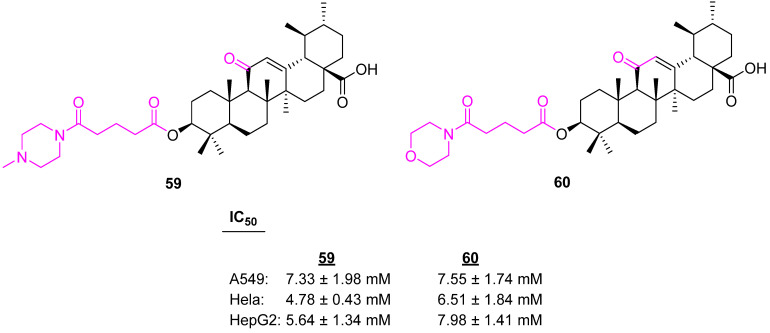
Structure of **59**, **60**.

**Table 1 molecules-27-08981-t001:** Summarized information on UA analogs (C3 modifications).

Compound	Pathway/Mechanism	Regulation	Cell Cycle Arrest (Phase)	Animal Study	[Ref]
**3**	Activation of caspases-8, -9, and -3	Apoptosis NA	Sub G1 & G2/M	no	[68]
**4**	Activation of p53	Bcl2↓ MDM2↓ Bax↑	Sub G1 & G1	no	[69]
**6**	NF−κB	NA	NA	no	[70]

**Table 2 molecules-27-08981-t002:** Summarized information on UA analogs (C28 modifications).

Compound	Pathway/Mechanism	Regulation	Cell Cycle Arrest (Phase)	Animal Study	[Ref]
**9**	Inhibition of extracellular signal-regulated kinase: ERK	NA	NA	no	[72]
**10**	Increased production of reactive oxygen species (ROS)	apoptosis	Sub G1	no	[73]
**12**	NA	NA	G2/M	no	[76]
**14**	DNA cleavage	NA	NA	no	[77]
**16**	Inhibition of HIF-1α transcriptional activity	HIF-1α↓ mRNA↓	G1	no	[79]
**17**	Mitotic catastrophe	NA	G2/M	no	[80]

**Table 3 molecules-27-08981-t003:** Concentrations of the half-maximal inhibition (IC50 ± SEM, μM) of compounds **21**–**25** on immortalized human fibroblasts, MCF7, U-87 MG, A549, HepG2 cells.

Compd.	Immortalized Human Fibroblasts	MCF7	U-87 MG	A549	HepG2
**Doxorubicin**	3.33 ± 0.67	4.51 ± 1.12	2.05 ± 0.22	6.17 ± 1.17	10.02 ± 1.67
**UA**	74.9 ± 4.58	74.9 ± 4.58	74.9 ± 4.58	74.9 ± 4.58	74.9 ± 4.58
**21a**	>100	>100	>100	>100	>100
**21b**	>100	>100	>100	>100	>100
**21c**	>100	>100	>100	>100	>100
**22a**	25.85 ± 3.04	22.9 ± 10.02	29.71 ± 6.52	>100	12.89 ± 2.63
**22b**	10.41 ± 1.25	1.55 ± 0.08	>100	>100	>100
**23a**	82.59 ± 16.87	26.23 ± 8.76	>100	>100	35.58 ± 8.83
**23b**	>100	>100	>100	>100	>100
**24a**	74.9 ± 4.58	>100	>100	40.26 ± 7.55	>100
**24b**	>100	>100	>100	>100	>100
**24c**	>100	>100	>100	>100	>100
**24d**	>100	>100	>100	>100	>100
**25**	48.35 ± 11.47	>100	81.67 ± 2.89	>100	>100

**Table 4 molecules-27-08981-t004:** IC_50_ values (µM) of compounds **43a**–**d** against various cell lines.

Compound	A549	MCF7	HTC116	THP1	FR2
**UA**	33 ± 0.03	37 ± 0.07	42 ± 0.08	9.1 ± 0.07	31 ± 0.08
**43a**	0.5 ± 0.05	5.5 ± 0.08	<0.1 ± 0.09	0.9 ± 0.02	10 ± 0.04
**43b**	2.9 ± 0.05	<0.1 ± 0.05	15 ± 0.06	<0.1 ± 0.03	69 ± 0.05
**43c**	<0.1 ± 0.001	<0.1 ± 0.09	0.3 ± 0.001	<0.1 ± 0.001	>50 ± 4.1
**43d**	0.15 ± 0.01	<0.1 ± 0.001	9.1 ± 0.1	<0.1 ± 0.001	>50 ± 3.9

**Table 5 molecules-27-08981-t005:** Summarized information on UA analogs (C3 and C28 modifications).

Compound	Pathway/Mechanism	Regulation	Cell Cycle Arrest (Phase)	Animal Study	[Ref]
**26**	Activation of caspase-3	Apoptosis	S	yes	[85]
**29**	Activation of the caspase-3 mitochondria pathway	Bcl2↓ Survivin↓	NA	yes	[89]
**31**	Activation of caspases-8, -9, and -3	Bcl2↓ HKII↓ Bax↑ p53↑	S G2/M	yes	[96]
**32**	NA	NA	G0/G1	no	[97]
**33**	Activation of caspases-8, -9, and -3 Increased production of reactive oxygen species (ROS)	Bcl2↓	G1	no	[98]
**36**	NF-κB	NA	G1	no	[100]

**Table 6 molecules-27-08981-t006:** Summarized information on UA analogs (modifications at multiple positions).

Compound	Pathway/Mechanism	Regulation	Cell Cycle Arrest (Phase)	Animal Study	[Ref]
**49**	NA	c-FLIP↓ NOXA↑ p21^waf1^↑	G1	no	[115]
**50**	NA	c-FLIP↓ NOXA↑ p21^waf1^↑	G1	no	[115]
**51**	Activation of caspases-8, -9, and -3		G1	no	[116]
**53**	Activation of caspase cascade RAS/Raf/MEK/ERK PI3K/AKT/mTOR Increased production of reactive oxygen species (ROS)	Bcl-2↓ Bax↑	S	no	[118]
**54**	NA	NA	G0/G1	no	
**55**	VEGF	HIF-1α↓	NA	no	
**56**	Caspase activation Increased production of reactive oxygen species (ROS)	Bcl2↓ Bax↑	G1	no	[124]
**57**	Caspase activation Increased production of reactive oxygen species (ROS)	Bcl2↓ Bax↑	G1	no	[124]
**58**	PI3K/AKT/mTOR Increased production of reactive oxygen species (ROS)	Bcl2↓ PARP↑ Bax↑ caspase-3/9↑	S	no	[125]

## Data Availability

Not applicable.

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
