# Peer review of "Ursolic Acid Analogs as Potential Therapeutics for Cancer"

_molecules, 2022, doi:10.3390/molecules27248981_

Round 1

Reviewer 1 Report

In general, the article has some practical usefulness, but it is not attractive and needs great improvement.

1. line 428-437: this paragraph is more appropriate as an introduction than as a conclusion.

2. The article lacks discussion: the article is a purely documentary enumeration, lacking the author's own summary and important views. To greatly attract readers, the author must examine in depth the functional variations of various UA derivatives and compare them.

3. The author should also summarize the molecular mechanism of each class of UA derivatives, like Figure 1, not just the observation of IC50.

4. line 55: add the description of UA sources appropriately, such as some described in Toxicon, 2019, 168: 141-146.

Author Response

Date:               December 12, 2022

Subject:           Revised Manuscript ID molecules-2102590 

Dear Reviewer,

Thank you for reviewing our manuscript. We appreciate your helpful and valuable suggestions. We have reviewed them carefully and believe we have made all appropriate changes to our manuscript.

Comments and Suggestions for Authors

In general, the article has some practical usefulness, but it is not attractive and needs great improvement.

Comment 1.

line 428-437: this paragraph is more appropriate as an introduction than as a conclusion..

Response: Thank you for your suggestion. We have now updated both the introduction and conclusion.

Comment 2.

The article lacks discussion: the article is a purely documentary enumeration, lacking theauthor's own summary and important views. To greatly attract readers, the author must examinein depth the functional variations of various UA derivatives and compare them.

Response: We have now discussed and summarized the information for better understanding.

Comment 3.

The author should also summarize the molecular mechanism of each class of UA derivatives,like Figure 1, not just the observation of IC50.

Response: We have now compiled the important piece of information in tabular form and briefly discussed them.

Comment 3.

line 55: add the description of UA sources appropriately, such as some described in Toxicon,2019, 168: 141-146.

Response: We have now included the source information as suggested by the reviewer.

Reviewer 2 Report

This review manuscript by Pandal et al. is of interest to people working in the Cancer research field. This review is well detailed and written, but would benefit if author had a broader view of what is described. So, this work would be strenghthened if author could describe a litte bit more the observations made in the referenced papers he cites, rather than only stating that compound X was used in A/B/C cancer. This is my first important comment. Second, in my opinion, but it's up to me to enhance clarity, it would have been great if authors displayed a table, each line corresponds to a modified compound with the associating effects. Finally, some references can be added in the paragraph starting from line 225 to 236: 

Kroemer G, Pouyssegur J. Tumor cell metabolism: cancer's Achilles' heel. Cancer Cell. 2008 Jun;13(6):472-82. doi: 10.1016/j.ccr.2008.05.005. PMID: 18538731.

Faubert B, Solmonson A, DeBerardinis RJ. Metabolic reprogramming and cancer progression. Science. 2020 Apr 10;368(6487):eaaw5473. doi: 10.1126/science.aaw5473. PMID: 32273439; PMCID: PMC7227780.

Pavlova NN, Zhu J, Thompson CB. The hallmarks of cancer metabolism: Still emerging. Cell Metab. 2022 Mar 1;34(3):355-377. doi: 10.1016/j.cmet.2022.01.007. Epub 2022 Feb 4. PMID: 35123658; PMCID: PMC8891094.

Cassim S, Vučetić M, Ždralević M, Pouyssegur J. Warburg and Beyond: The Power of Mitochondrial Metabolism to Collaborate or Replace Fermentative Glycolysis in Cancer. Cancers (Basel). 2020 Apr 30;12(5):1119. doi: 10.3390/cancers12051119. PMID: 32365833; PMCID: PMC7281550.

Cassim S, Raymond VA, Lacoste B, Lapierre P, Bilodeau M. Metabolite profiling identifies a signature of tumorigenicity in hepatocellular carcinoma. Oncotarget. 2018 Jun 1;9(42):26868-26883. doi: 10.18632/oncotarget.25525. PMID: 29928490; PMCID: PMC6003570.

Author Response

Date:               December 12, 2022

Subject:           Revised Manuscript ID molecules-2102590 

Dear Reviewer,

Thank you for reviewing our manuscript. We appreciate your helpful and valuable suggestions. We have reviewed them carefully and believe we have made all appropriate changes to our manuscript.

Comments and Suggestions for Authors

This review manuscript by Pandal et al. is of interest to people working in the Cancer researchfield. This review is well detailed and written, but would benefit if author had a broader view ofwhat is described. So, this work would be strenghthened if author could describe a litte bit morethe observations made in the referenced papers he cites, rather than only stating that compoundX was used in A/B/C cancer. This is my first important comment. Second, in my opinion, but it'sup to me to enhance clarity, it would have been great if authors displayed a table, each linecorresponds to a modified compound with the associating effects. Finally, some references canbe added in the paragraph starting from line 225 to 236:

Kroemer G, Pouyssegur J. Tumor cell metabolism: cancer's Achilles' heel. Cancer Cell. 2008Jun;13(6):472-82. doi: 10.1016/j.ccr.2008.05.005. PMID: 18538731.

Faubert B, Solmonson A, DeBerardinis RJ. Metabolic reprogramming and cancer progression.Science. 2020 Apr 10;368(6487):eaaw5473. doi: 10.1126/science.aaw5473. PMID: 32273439;PMCID: PMC7227780.

Pavlova NN, Zhu J, Thompson CB. The hallmarks of cancer metabolism: Still emerging. CellMetab. 2022 Mar 1;34(3):355-377. doi: 10.1016/j.cmet.2022.01.007. Epub 2022 Feb 4. PMID:35123658; PMCID: PMC8891094.

Cassim S, Vučetić M, Ždralević M, Pouyssegur J. Warburg and Beyond: The Power of Mitochondrial Metabolism to Collaborate or Replace Fermentative Glycolysis in Cancer. Cancers(Basel). 2020 Apr 30;12(5):1119. doi: 10.3390/cancers12051119. PMID: 32365833; PMCID:PMC7281550.

Cassim S, Raymond VA, Lacoste B, Lapierre P, Bilodeau M. Metabolite profiling identifies a signature of tumorigenicity in hepatocellular carcinoma. Oncotarget. 2018 Jun 1;9(42):26868-26883. doi: 10.18632/oncotarget.25525. PMID: 29928490; PMCID: PMC6003570.

Response: Thank you for your suggestion. We have now carefully reviewed updated the manuscript and included the suggested references.
